# Flow Dynamics through a High Swelling Nanofiber Membrane Processed at Different Relative Humidities: A Study on a FexOy/Polyvinyl Alcohol Composite

**DOI:** 10.3390/membranes14090189

**Published:** 2024-08-30

**Authors:** Ayelen C. Santos, Alicia Vergara-Rubio, Angel J. Mazocca, Silvia Goyanes

**Affiliations:** Laboratorio de Polímeros y Materiales Compuestos, Instituto de Física de Buenos Aires—CONICET (IFIBA), Universidad de Buenos Aires, Buenos Aires C1428EHA, Argentina; asantos@df.uba.ar (A.C.S.); avergarar@df.uba.ar (A.V.-R.); marzo@df.uba.ar (A.J.M.)

**Keywords:** electrospinning, swelling, flow dynamics, nanocomposite

## Abstract

Addressing the global problem of polluted water requires sustainable, efficient, and scalable remediation solutions, such as electrospun polyvinyl alcohol (PVA) membranes incorporating specific nanoadsorbents. The retention of contaminants depends on membrane swelling, morphology, and the adsorbent within the nanofiber. This study investigated the effect of relative humidity (RH) within the electrospinning chamber on the morphology of the resulting mats and how this affected the flow dynamics depending on whether or not the permeating liquid induced swelling in the membranes. An insolubilized PVA membrane was used as a hydrophilic filter model and a PVA membrane filled with iron oxide nanoparticles (IONPs) as a composite model (PVA + IONPs). The presence of IONPs increases the nanofiber diameter, which decreases when prepared under intermediate RH (IRH). Consequently, the nanofiber configuration, which is critical for filtration tortuosity, is influenced by RH. The initial swelling results in over 60% greater water flux through PVA + IONPs compared to PVA at an equivalent RH. This characterization helps to optimize membrane applications, highlighting that PVA + IONPs exhibit lower permeability values at IRH, indicating improved contaminant retention capabilities.

## 1. Introduction

There has been growing interest in exploring electrospinning as a method for producing micro/nanoporous membranes for water treatment [1]. Polymers such as polyacrylonitrile or polybutylene adipate terephthalate are typically used for this application [2,3,4,5]. However, these polymers require dissolution in solvents like acetone or dimethyl sulfoxide, which are neither environmentally friendly nor conducive to operator safety. In response to this challenge, polyvinyl alcohol (PVA) is considered a green polymer and it has emerged as a promising alternative because it is electrospinnable using water as a solvent, avoiding the use of organic solvents [6,7,8].

The electrospun PVA mat is inherently water soluble, but it is well known that this issue can be resolved through thermal treatment or the use of crosslinking agents [9,10]. Torasso et al. [11,12] demonstrated that applying only a short heat treatment (HT) at 190 °C for 15 min allows the PVA electrospun mat with and without IONPs content to become insoluble. Despite maintaining their hydrophilic nature, PVA electrospun mats with short HT maintain a high swelling capacity in water [12,13]. Swelling produces strong changes in the membrane morphology. As the fibers absorb water, their diameter increases and the pore size decreases until the equilibrium configuration is reached. This continuous process, which occurs during the first few minutes of filtration, has two consequences: the desired effect of retaining smaller contaminants and the undesired reduction in the permeate flow rate. While this point has been acknowledged [13,14], it remains unexplored in-depth. Understanding this aspect is crucial for the practical application of such membranes.

Organic/inorganic electrospun membranes were utilized for antibacterial treatment or the removal of ionic contaminants, which can be retained by nanoadsorbents contained in the nanofibers [15,16,17,18,19,20,21]. Several PVA nanocomposites with various nanoadsorbents were utilized for the removal of multiple contaminants, including pharmaceuticals and Pb(II), Cr(VI), Cu(II), As(V), and As(III), achieving high levels of efficiency [14,22]. However, all of these studies were conducted under batch conditions rather than using a continuous approach, which is more applicable to their potential use in a water filtration system [7].

There are several studies on the filtration dynamics of various polymeric nanocomposite membranes, including PVA, but they are either not obtained using the electrospinning technique or do not address polymer swelling [23,24]. The literature lacks precedent for studying swelling and the associated changes in flow dynamics in electrospun PVA nanocomposites.

Electrospun PVA mats incorporating iron oxide nanoparticles (IONPs) or iron ions have recently been recognized for their remarkable arsenic removal efficiency under batch conditions [11,12,25]. However, since the potential use of this material as a filtration membrane is particularly interesting, critical properties such as swelling and water flow dynamics have yet to be studied.

Another highly relevant, often overlooked, aspect when electrospinning hydrophilic polymers, is the control of relative humidity (RH) within the chamber where electrospinning takes place. It is well known that the RH of the electrospinning chamber is a fundamental factor in the process that determines the final morphology of the material when hydrophilic polymers are used [26,27]. In particular, the effect on the nanofiber diameter of PVA electrospun membranes with different RH conditions has been reported in several works, leading to wider nanofiber when RH decreases [26,27]. In the case of PVA blends, as far as we know, there is limited research on how the RH condition in the electrospinning chamber affects the final membrane morphology [26,28,29]. However, to our knowledge, there is no precedent in the literature for studying the effect of RH on PVA nanocomposites.

Besides, the relation between RH and the swelling property of PVA nanocomposites has not been explored until this research. Although swelling, water, and oil/water filtration have been studied for PVA nanocomposite membranes, particularly in the context of water remediation and oil separation [30,31], the effect of RH has not been investigated. Thus, there is a significant gap in understanding the correlations between the humidity at which PVA nanocomposites are processed and its impact on morphology, swelling, and flow dynamics.

Moreover, none of the previous studies specifically focus on integrating swelling and permeation dynamics with a hydrophilic polymer, or on how these factors relate to the morphology under different RH conditions during fabrication. Understanding how the swelling mechanism is influenced by manufacturing conditions and the presence or absence of IONPs, as well as its impact on performance during permeation, is crucial for the potential applications of this material.

The focus of this study is to understand the morphological changes experienced by electrospun membranes made from a hydrophilic polymer and nanocomposites of the same polymer, depending on the RH during fabrication. We investigated how both types of membranes are affected by water swelling and explored how these morphological alterations impact the dynamic permeate flow through them. An insolubilized PVA electrospun membrane by heat treatment was employed as the hydrophilic polymer model, while PVA containing IONPs served as the hydrophilic nanocomposite model (PVA + IONPs). Permeate flow curves over time were analyzed in relation to the RH at which the membranes were manufactured, along with statistical parameters such as fiber diameter distributions, mesh tortuosity, swelling, porosity, and permeability.

## 2. Materials and Methods

### 2.1. Materials

PVA-Mowiol 10-98 with a molecular weight of 61,000 g/mol and a hydrolysis degree of 98%, NaOH, FeCl_3_ were obtained from Sigma Aldrich^®^ in Buenos Aires, Argentina. FeSO_4_⋅7H_2_O salts were purchased in Biopack (Buenos Aires, Argentina). All reagent grades were used for IONPs synthesis without any additional purification.

### 2.2. Electrospinning Solutions and Process

Two types of membranes were prepared using an aqueous PVA solution (12 wt%) with and without 1% IONPs (wt% with respect to PVA), following methodologies previously reported in [9,12]. Briefly, in the case of PVA + IONPs, a mixture of two iron salts (FeCl_3_ y FeSO_4_·7H_2_O) was added to an aqueous PVA solution. Then, the IONPs were formed in situ by adding NaOH 1 M by drip. This 1% was chosen because a previous work showed a good dispersion with this value and higher concentrations lead to an increase in their agglomeration [12,32]. It is also known that a higher concentration of inorganic nanoparticles means a higher viscosity, leading to the worst dispersion [32].

The relative humidity of the electrospinning chamber during membrane production was intermediate (IRH: 55 ± 5% RH) and low (LRH: 35 ± 5% RH). For both relative humidity conditions, PVA and PVA + IONPs in homogeneous aqueous solutions were used for membrane preparation. The viscosity of the solutions was measured using a Brookfield viscometer model LV DV-E™ (AMETEK Brookfield, Marlborough, MA, USA) with an S31 spindle at 50 rpm. Conductivity and pH measurements were performed with an Orion™ Versa Star Pro™ (Thermo Fisher Scientific, Waltham, Mass., USA). Surface tension measurements of both solutions were performed using an Attension Theta Lite Tensiometer (Biolin Scientific, Västra Frölunda, Sweden) at room temperature by adding a drop of solution (~13 μL). The average value of five measurements was reported. The relevant physical properties of the electrospun solutions are listed in Table 1.

#### Preparation of Electrospun Membranes

Aqueous polymer solutions contained in 20 mL syringes were electrospun in a TL-Pro-BM robotic electrospinning machine (Shenzhen, China) with an RH controller consisting of an external dehumidifier and an internal humidifier. The process was maintained at two different RH conditions, 35 ± 5% (LRH) and 55 ± 5% (IRH), at 25 °C with a 2 mL·h^−1^ controlled feed rate through a multi-needle device with five 21G needle (0.9 mm inner diameter). Two high-voltage power supplies were used, applying a voltage of 22 kV to the needle and −8 kV to the collector. The collector is a drum rotating at 300 rpm with a diameter of 10 cm, covered with a cleaned stainless steel mesh, and positioned 12.5 cm from the needle tips. After 9 h of electrospinning, a mat (65 ± 5) μm thick was obtained. These membranes are then heat treated at 190 °C for 15 min to achieve water stability by the dehydration of OH. In previous work, it has been shown that a treated membrane (PVA) maintains its integrity when immersed in water [9,13]. To simplify the nomenclature, from here on, we will refer to the insolubilized electrospun PVA membrane simply as PVA.

### 2.3. Morphology and Structure of Mats

To determine the influence of RH during the manufacturing process, the membrane morphology was observed by field emission SEM using FEG-SEM, FEI-QUANTATM 250 (Hillsboro, Ore., USA). From the obtained images, the nanofiber diameter distribution of each mat was determined using an automated method with ImageJ software (1.51w, Wayne Rasband, National Institutes of Health, Stapleton, N.Y., USA), as reported in the literature [33,34,35]. To analyze the nanofiber density of the material, samples of each membrane type were cryogenically fractured in liquid N_2_. Their nanofiber entanglement structure was also examined by field emission SE.

In the case of PVA + IONPs electrospun mats, a complementary study was made by Transmission Electron Microscopy (MET Zeiss 109, Oberkochen, Germany) in Appendix A, Fourier Transform Infrared Spectroscopy (Spectrum Two Perkin Elmer, Shelton, CT, USA) in Appendix A and X-ray diffraction (Philips PW3710, Almelo, The Netherlands), see Appendix A, in order to verify the IONPs formation, size, and distribution within the nanofibers, see Appendix A.

### 2.4. Porosity Measurements and Effective Pore Size Estimation

The Liquid Intrusion Method (LIM) was used to evaluate the porosity of the scaffold [36]. The electrospun scaffolds were first weighed, then immersed in a lubricant (which does not cause swelling in the membrane; in this case, it was Vaseline) overnight at room temperature in a beaker placed on a mechanical shaker to facilitate the infiltration of the lubricant into the voids of the scaffold. Following this, the exterior of the specimens was carefully blotted to remove excess moisture and then reweighed to determine the mass of lubricant that had permeated the scaffold structure. This procedure was repeated five times for each scaffold variant. Porosity (*V_u_*) was subsequently calculated using the following formula:(1)$$V_{u}=\frac{V_{V}}{V_{V}+V_{Pol}}$$
where *V_v_* represents the volume of infiltrated Vaseline, indicating the total pore volume. This is calculated by determining the ratio of the observed mass change after infiltration to the density of the lubricant (ρ_V_ = 0.87 g·cm^−3^). *V_pol_* is the volume of the polymer, obtained as the ratio of the initial mass of the dry scaffold before liquid infiltration to the density of the polymer (ρ_pol_).

In a highly simplified model, if the membrane is considered a continuous material with a single cylindrical pore through which the permeating fluid flows, its size would be determined by the classical Poiseuille equation [37]. However, electrospun membranes are more complex structures where tortuosity, influenced by nanofiber configuration and material swelling, governs fluid flow through the membrane. Therefore, a simplified model can be applied taking into account a homogeneous distribution of spherical voids with an effective radius (*r_eff_*), the number of which is determined by the porosity of the membrane *V_u_*. A possible approach for these materials is to consider a modified version of the classical Poiseuille’s law for permeate flow through a medium with a homogeneous distribution of spherical holes with radium *r_eff_*. The permeate flow *J*, of a liquid of viscosity *η*, driven by a pressure difference *∆P*, across a membrane with thickness *L*, and porosity *V_u_*, is given by Equation (2):(2)$$J=V_{u}\frac{r_{eff}^{2}}{8\eta }\frac{\Delta P}{L}$$

Nevertheless, this model warrants careful experimental examination, as the permeating liquid through the membrane must be chosen to avoid altering its structure. In the case of our materials, the solvent must be selected to prevent the swelling of the PVA. Therefore, as described in Section 2.5, we investigated membrane swelling with toluene and found negligible swelling for all four materials presented.

### 2.5. Swelling of the Mats

To evaluate the swelling of all materials and its dependence on the RH condition during fabrication, swelling tests were conducted. Three samples of each material were weighed and then immersed in distilled water and toluene for 72 h. Subsequently, the samples were placed on blotting paper for 1 s to remove the surface solvent from both sides, and their mass was immediately measured. This procedure allowed for the determination of the amount of solvent retained within the material.

Swelling estimations can be made measuring the amount of solvent moles absorbed by 100 g of membranes, *Q*. They were made following the relation [38]:(3)$$Q=\frac{\frac{m_{ss}}{M_{solv}}}{m_{pol}}\times 100$$
where *m_ss_* is the absorbed solvent mass, *m_pol_* is the polymer mass, and *M_solv_* is the molar mass of the solvent. Toluene measurements did not cause the mats to swell, as the mass before and after immersion was the same in all cases.

Swelling is the main driver of the flow dynamic effects of the membranes, which is highly dependent on the water absorption capacity of the nanofiber material. Therefore, it is crucial to determine the wetting capacity of PVA alone and with the IONPs. Therefore, we prepared continuous films of PVA and PVA + IONPs and measured the contact water across them. This interaction analysis between each material of the nanofibers and the two solvents used was also studied by contact angle (*θ*) and physisorption time measurements, performed using an Attension Theta Lite Tensiometer (Biolin Scientific, Västra Frölunda, Sweden). These tests were performed at room temperature by dropping a drop of water or toluene (~6 μL) on the surface of a continuum film of each solution. The average value of five measurements was reported. In the case of the physisorption of the materials, the reported time corresponds to the moment where *θ* reaches 0°.

### 2.6. Flow through the Mats with Pure Water and Toluene

The permeate flow *J* was obtained from the measurement of the collected permeate volume *V* over time *t*. It was calculated using the following equation: $J=\frac{\Delta V}{A\Delta t}$, where *A* is the membrane surface area.

In this research, we analyzed the filtration dynamics using two different solvents: toluene and distilled water (pure water). To assess the filtration dynamics of the material without fiber swelling, tests were performed with toluene (Biopack^®^, Buenos Aires, Argentina) to ensure that the pore size remained constant during filtration. 

Disks with a diameter of (25.0 ± 0.1) mm were cut from each membrane using a die. Their mass and thickness (*L*) were measured, and then they were inserted into a filter holder (Figure 1). The electrospun fibrous membrane was placed between two plastic rings, leaving a circular effective permeation area of (3.0 ± 0.1) cm^2^. A precision balance was used to measure the mass of solvent filtered through the membranes over time at a pressure of (4.0 ± 0.5) mbar at room temperature (25 °C). Narrow-mouth containers were used to minimize solvent evaporation during the experiments. 

The permeation tests for the *J* measurements were conducted using a system that maintains constant hydrostatic pressure on the membrane while measuring the weight of permeated liquid. The device (Figure 1) consists of a graduated acrylic cylinder, with the filter holder with the membrane located at its base. The liquid (in our tests, toluene or distilled water) is placed in a reservoir connected to the cylinder. As the level of the liquid in the cylinder remains constant, its pressure on the sample is also constant when the liquid passes through the membrane. This is achieved by using the pump to refill the internal reservoir from an external one until it overflows, ensuring a consistent water level.

As described in Section 2.4, Equation (2) suggests that the decrease in flow is a consequence of the reduction in the effective pore radius (*r_eff_*), caused by the entanglement of the nanofibers. However, it is important to note that the nanofiber network should not be thought of as consisting of continuous “holes”. The tortuosity of the fiber structure causes the fluid to deflect and pass through pores of varying radii and lengths. Therefore, Darcy [39] retained the proportionality between pressure and flow and introduced a new variable for the transport of Newtonian fluids through porous media, known as intrinsic permeability (*κ*), which encapsulates information about pore structure and tortuosity. Assuming the membrane is isotropic and κ is a scalar, Darcy’s law is expressed as follows:(4)$$J=\frac{-\kappa }{\eta }\frac{\Delta P}{L}$$

In the description of Darcy, the experimental results indicate that κ depends on time and the type of fluid passing through the membrane (whether it causes swelling or not). If there is no interaction between the solvent and the membrane, and thus no swelling, the pore size remains constant during the flow process. Therefore, permeability can be estimated from the permeate flow.

## 3. Results and Discussion

### 3.1. Morphological Characterization of Electrospun Mats (PVA and PVA + IONPs). Effect of Chamber RH

Figure 2a,b show SEM micrographs of PVA electrospun membranes under the two RH conditions studied. In both cases, the nanofibers are entangled and randomly oriented in a plane parallel to the substrate. However, their diameter (ϕ) distribution is clearly different (Figure 2c). The statistical parameters (mode, mean and standard deviation) are shown in Table 2.

RH has a significant influence on the nanofiber diameter distribution (See Figure 2c and Table 2). At intermediate RH, all of the statistical parameters of the distributions show a marked decrease.

This result is consistent with the trend shown by Pelipenko et al. [26] and with data from the literature [12,27,29,40] obtained at different RH (Figure 3). In our case, the ϕ_mean_ decreases by 50% as RH increases. Although the diameters obtained in our work are slightly smaller than those reported for the same RH, this discrepancy may be due to differences in process conditions, molecular weight, and the hydrolysis degree of the PVA used.

In the electrospinning process, RH conditions affect the evaporation rate of the solvent from the polymer solution: higher RH conditions lead to lower evaporation rates, which result in a slower solidification of the polymer jet as it moves from the spinneret to the collector [26,41,42]. This extended solidification time allows for an increased stretching and elongation of the polymer jet, producing thinner nanofibers with reduced diameters. Conversely, lower RH conditions accelerate the evaporation rate, resulting in a faster solidification of the polymer jet and thicker nanofibers with larger diameters.

In the case of PVA + IONPs membranes, the IONP presence and distribution inside the nanofibers are shown in the Appendix A. The nanoparticles in the solution to be electrospun strongly influence the effect of the chamber RH on the morphology of the membranes obtained. Figure 4a,b show SEM micrographs of PVA + IONPs membranes under different RH conditions. Figure 4c and Table 2 show the ϕ distribution for these mats and their statistical parameters.

The results show that a symmetrical ϕ distribution is not achieved under any process condition (Table 2 and Figure 2c and Figure 4c). This is evident from the notable differences in the mode and mean values of each distribution. Additionally, the effect of chamber humidity on ϕ is much more pronounced for the PVA mat than for the PVA + IONPs mat. A noteworthy observation is that the diameter distribution for the PVA + IONPs membrane, when processed under LRH, exhibits a bimodal shape, which will be discussed later.

In the case of materials with nanoparticles, there are no reports on the influence of the humidity at which the mat is produced. However, the results presented in this paper can be understood by considering that nanoparticles modify the transport properties of the solution being electrospun. In particular, they affect the evaporation and sorption of water during the process. Pelipenko et al. [26] show that PVA blends with less hydrophilic polymers reduce the effect of chamber RH on nanofiber diameter. Raksa et al. [28] studied a blend of silk fibroin (SF)/PVA and observed a slight variation in the nanofiber diameter, with a decrease in the mean diameter of 5% as the RH changes from 50% to 70%. In our PVA + IONPs samples, there is a 38% decrease in the ϕ_mean_ of the nanofibers as the RH of the manufacturing process changes from 35% to 55%, following the trend shown for nanofibers made of PVA alone. However, it should be emphasized that the variation in ϕ_mode_ for the same change in RH process conditions, is 57% greater in the case of PVA membranes.

Comparing the ϕ_mean_ of PVA membranes with those of PVA + IONPs reveals additional differences. The shape of the ϕ distribution for the PVA + IONPs mat produced under LRH conditions (Figure 4c) shows a bimodal diameter distribution. In Figure 5, these two distributions are separated, each of which is assumed to be Gaussian, and their ϕ_mean_ and standard deviations are given in Table 2. Therefore, when using LRH during the production of PVA + IONPs membranes, a larger proportion of nanofibers with smaller diameters was observed.

Our results show a change in the morphology of the membranes due to the presence of IONPs. This can be explained by the competition between two mechanisms operating during electrospinning. Changes in conductivity affect the polarization of the solution due to the presence of the electrical field, which strongly alters the nanofiber size [8,43]. Moreover, changes in viscosity also affect the elongation of the nanofiber during the electrospinning process, making it a crucial factor in determining the final size of the nanofiber [15,21,44].

As can be seen in Table 1, the presence of IONPs effectively generates significant differences in both conductivity and viscosity. Haghi et al. [44] show that the mean nanofiber diameter results from a competition between the applied electric field and the viscosity of the solution. Varying the conductivity of the solution while maintaining a fixed electric field is equivalent to varying the electric field while keeping the conductivity constant. Therefore, increasing the conductivity has the same effect as increasing the electric field. Moreover, an increase in the electric field decreases the nanofiber diameter, which is also observed when conductivity is increased. Conversely, as the viscosity of the solution increases, the nanofiber diameter of the membrane also increases. This indicates that increasing either the conductivity or the viscosity of the solution leads to competing effects on the nanofiber diameter of the membrane.

In addition, Ruixiang Xu et al. demonstrated that in PVDF membranes, a simultaneous increase in the conductivity and viscosity of the electrospinning solution led to an increase in the mean nanofiber diameter [15]. On the other hand, Byung Wook Ahn and Tae Jin Kang studied the effects of incorporating various concentrations of iron oxide nanoparticles into the electrospinning mixture on the morphology of the resulting fibers [32]. They found that the average diameter of the poly (ethylene terephthalate)/iron oxide nanofibers increased with nanoparticle concentration. While these nanoparticles acted as charge carriers, thereby increasing the Coulomb force, they also influenced the viscosity of the solution, which increased proportionally with the iron oxide content. The researchers concluded that the effect of viscosity predominated over the increase in Coulomb force and that, as nanoparticle loading increased, aggregates tended to form.

In our case, the conductivity of the PVA + IONPs solution is 42% higher than that of the PVA solution (Table 1), which benefits jet elongation in an electrostatic field. The higher net charge density increases the electrical force exerted on the jet and leads to a decreased ϕ [45]. On the other hand, the viscosity of the PVA + IONPs solution is 184% higher than that of the PVA solution (Table 1) resulting in an opposite effect compared to conductivity, thereby increasing ϕ. The composite solution exhibited high viscosity, while the surface tension remained virtually unchanged (Table 1). Consequently, the stretching ability was reduced. The charged jet of the high-viscosity solution could withstand the Coulombic stretching force, resulting in larger nanofibers [46]. Additionally, the low solvent content in the charged jet of the PVA + IONPs solution dried more easily, limiting the stretching and thinning of the nanofibers. This effect is particularly pronounced in low-humidity processes, where the evaporation rate is faster. This could explain the bimodal distribution observed for LRH (Figure 5). In contrast, the slower evaporation rate in high-humidity processes increases the process time and allows for a more homogeneous distribution of ϕ (narrower distribution, Figure 4c).

The presence of the IONPs within the nanofibers creates an interface with the polymer matrix [12]. This reduces the impact of the stretching effect on the nanofiber due to the RH of the electrospinning chamber, an effect observed in the obtained values of ϕ_mean_ given in Table 2.

Determining the pore distribution between nanofibers is another way to assess the effect of RH during membrane production. This was achieved by analyzing SEM images; the pore area distribution between the nanofibers was obtained from data analysis using DiameterJ® software (v1.018, Rostock, Germany) [33,34,35] (Figure 6). For both materials, there are more smaller pores under LRH conditions than under IRH. The insets of Figure 6 show that for membranes produced under IRH conditions, the number of pores decreases smoothly as the pore area increases. In contrast, membranes produced under LRH conditions exhibit a more random distribution of pore areas, regardless of whether the membrane is PVA or PVA + IONPs.

The obtained pore area distribution is based on a 2D image. This distribution may vary through the thickness of the mat. Therefore, this analysis should be taken as a preliminary approach. To fully understand the permeation dynamics, tortuosity, and porosity, which are 3D properties, further analysis is required.

Figure 7 illustrates the calculated porosity of all membranes using Equation (1), with the highest value observed in the PVA + IONPs under LRH conditions. Notably, this is the only membrane where the ϕ distribution exhibited a bimodal function (Figure 5). To evaluate tortuosity, SEM images of the fracture surface of each mat were obtained (Figure 8). For both PVA and PVA + IONPs membranes, chamber humidity has a clear effect on fiber entanglement morphology. Figure 8a,b show that LRH produces a less dense material compared to IRH, resulting in lower tortuosity. For PVA + IONPs membranes, Figure 8c,d, LRH appears less compact than IRH, suggesting lower tortuosity for membranes produced under LRH conditions.

These differences in material morphology directly impact the filtration behavior of the membranes, due to alterations in the available spaces through which a fluid can traverse. Changes can be observed in the effective pore radius (*r_eff_*). In order to obtain this value for each membrane, toluene was used as a solvent for flow measurements without causing swelling. Toluene, an organic solvent, does not induce swelling in hydrophilic polymers like PVA [47].

The dynamic flow of toluene through electrospun membranes does not show significant variations in permeation time as expected. Figure 9 shows the average values of toluene permeate flow, normalized to thickness (*J*·Δ*Z*), for PVA and PVA + IONPs membranes produced under different RH conditions. The permeate flow of toluene through membranes fabricated under LRH conditions is more than 30% higher than that through membranes fabricated under IRH conditions. For toluene, the contact angles measured on continuous films for both materials (to evaluate only the effect of the material and avoid the effect of the nanostructure) are (8.1 ± 2.4)° for PVA and (13.8 ± 2.6)° for PVA + IONPs.

To analyze these results, it is important to remember that in the electrospinning process, the injection rate of the solution remains constant across all cases. Therefore, a reduction in ϕ would be expected to result in a greater number of fibers due to the conservation of injected mass. This should indicate that the mat tortuosity is higher under IRH conditions compared to LRH conditions, as suggested by the images in Figure 8.

From the porosity and toluene filtration measurements for each membrane, *r_eff_* was estimated using Equation (2), with results shown in Figure 10. These results allow us to conclude that under IRH conditions there are no differences in *r_eff_* between the pure PVA membrane and the nanocomposite one. However, when produced under LRH conditions, *r_eff_* is higher in the PVA + IONPs membrane.

The application of these membranes is for water remediation and since the polymer used is PVA, a swelling effect will occur. For this reason, it is necessary to study the swelling that occurs for each of the obtained morphologies and then determine how this affects the permeation dynamics.

### 3.2. Swelling: Interaction Solvent Materials

When a solvent that causes swelling in the material passes through the membrane, this swelling introduces an additional effect on the filtration process. For pure water and PVA, it is well known [13] that water flow decreases over time due to the nanofiber swelling. For our membranes, the solvent uptake in pure water by 100 g of membrane, *Q* was calculated using Equation (3) and is given in Figure 11. 

It is interesting to note that membranes with larger mean nanofiber diameters result in lower *Q*. There appears to be a correlation where membranes with smaller diameter fibers exhibit higher *Q*, regardless of whether this is achieved by increasing RH or adding nanoparticles.

However, analyzing the swelling process in nanofiber membranes requires careful consideration. The experiment involves immersing the membrane in water and allowing it to swell until a constant mass of adsorbed water is reached. While some of the water will be adsorbed due to the swelling of the nanofibers, some will remain as occluded water in the nanopores of the membrane, even after both surfaces are dried as part of the experimental protocol.

### 3.3. Dynamic Flow through the Membranes with Water

Figure 12a,b show the time dependence of the permeated pure water flow through the different membranes studied. In all cases, the permeate flow decreases continuously until it reaches a steady state.

The first 60 min of Figure 12a,b are shown in detail in Figure 13a,b.

The initial flow through the PVA + IONPs membrane (Figure 13b) is at least 60% greater than that through the PVA membrane (Figure 13a) produced under the same RH conditions. This difference may be related to the hydrophilicity of the materials reflected in their contact angle (*θ*) measurements. *θ* on PVA continuous films, measured after 5 s, was (35.3 ± 6.0)°, and on PVA + IONPs films it was (55.6 ± 4.3)°. With a higher contact angle, droplets formed on the nanofibers have a larger diameter, which reduces the restriction of water flow. In addition, the physisorption time (TP) with water for the composites is longer than that for PVA alone (TP_PVA_ = (957 ± 28) s and TP_PVA + IONPs_ = (1519 ± 24) s). 

The behavior of *J*Δ*Z* curves (Figure 12) is due to the swelling of PVA. When it reaches its maximum water absorption capacity, the *r_eff_* of the membrane takes its minimum value and remains constant. This phenomenon has been reported in previous studies for PVA and is consistent with our findings. It can be seen that within a period of just over a day (1700 min), the flow rate remains constant after reaching the plateau. Consequently, the pore size does not change and the membranes behave as a filter with a fixed pore size over time, specifically in the case of pure water. However, in the case of contaminated water, the pore size will reduce due to the known fouling effect until the point of occlusion. This behavior has been demonstrated in PVA electrospun mats that were used for the filtration of TiO₂ nanoparticles in water [13].

Comparing Figure 12a,b, it can be concluded that the presence of IONPs inside PVA nanofiber limits the swelling behavior of the membrane, i.e., the permeate flow is higher than PVA alone.

Figure 14 illustrates the behavior of a pore when the nanofiber of the PVA membrane interacts with different solvents, such as toluene (Figure 14a) or water (Figure 14b,c). When the solvent is toluene, no swelling occurs in the nanofibers. However, the materials swell when interacting with water. The kinetics of swelling has an associated rate. Initially, as water passes through the membrane, the swelling is minimal due to the initial fiber diameter (ϕ_1_) which is the smallest possible value. As the process continues, the average pore size between the fibers decreases because the fiber diameter increases (ϕ_2_) until an equilibrium state is reached.

On the other hand, the initial water flow for all membranes (Figure 13) differs from the toluene measured flow (Figure 9) by at least 40% for PVA membranes and less than 6% for PVA + IONPs membranes. This variation may be attributed to the altered hydrophilicity of PVA in the presence of IONPs, which leads to a decrease in flow compared to PVA without IONPs. Additionally, differences in values may also arise from water being retained in the pores, as illustrated in the blue area between the nanofiber and the pore in Figure 14. Toluene flows through the membranes more easily than water.

PVA membranes are porous media. To analyze the dynamic behavior of the flow, it is necessary to determine the permeability *κ*. This can be obtained from Equation (4) and its values are shown in Figure 15.

For toluene filtrations, *JΔZ* remains constant, so *κ* is also constant (Figure 15a). However, for water filtration, *JΔZ* changes during the permeation test (Figure 12a,b) making *κ* a function of time. Therefore, *κ* was evaluated at the initial and final states (steady state) with the corresponding values given in Figure 15b,c.

The initial permeability of the obtained materials with different fluids (Figure 15a,b) correlates with the intrinsic tortuosity of each material (Figure 8). A lower tortuosity or a less dense nanofiber structure results in a higher *κ* value. This means that *κ* is strongly dependent on the RH conditions applied during the electrospinning process used to fabricate the membrane. 

Both PVA and PVA + IONPs membranes fabricated under IRH conditions show lower permeability compared to those fabricated under LRH conditions. The reduced permeability of IRH membranes (Figure 15) is due to a denser structure formed during the membrane fabrication process (as seen in Figure 8b,d), which restricts water flow through the membrane. 

Comparing *κ* for the different fluids interacting with the membranes, as seen in Figure 15a,b, the values for PVA + IONPs with distilled water are higher than those for toluene.

As mentioned above, our contact angle measurements for toluene/PVA and toluene/PVA + IONPs were larger than those obtained for water/PVA and water/PVA + IONPs. Furthermore, toluene is not absorbed by these materials, whereas they absorb water in a relatively short time.

On the other hand, the *κ* values for PVA at LRH and PVA at IRH with distilled water (Figure 15b) show no significant differences, whereas for PVA + IONPs, there are observable differences. All of these outcomes align with expectations when analyzing flow measurements from individual membranes, as illustrated in Figure 13. 

In Figure 15b, we found that the incorporation of IONPs into PVA membranes significantly enhances permeability, likely due to the increased pore size and reduced tortuosity resulting from the presence of IONPs. Although the LRH membranes show lower water flow permeation than the IRH membranes, from the perspective of arsenic remediation, it is better to maximize the contact time of the contaminated water with the fibers containing the adsorbent in their structure. This relation is also maintained in the final *κ* at steady state, Figure 15c, where it reduces significantly. This is important because even during prolonged filtration, the PVA + IONPs membrane under IRH conditions retains a lower *κ*, which enhances its performance as both a filtration membrane and an adsorption material.

Finally, the analysis of the steady-state conditions (where maximum fiber swelling is achieved) allows for the characterization of both the effective pore radius (*r_eff_*) and the permeability *κ* of the material, based on the same measurements and using Equations (2) and (4), respectively. Figure 15c presents the obtained *κ* values, revealing similar trends that align with the initial tortuosity of all materials. Furthermore, a significant decrease in the absolute *κ* values of all materials can be observed when compared to the initial membrane permeability during filtration (Figure 15b). This reduction is expected due to the swelling effect of PVA and the decrease in effective pore size through which water can pass as the nanofibers swell, in contrast to the initial state (refer to the schematic in Figure 14). It can be discerned that to minimize the permeability of the PVA + IONPs material, it should be fabricated under conditions of intermediate or high RH. On the other hand, regarding the effective pore radius, Figure 16 shows the *r_eff_* obtained at the final state of swelling.

As expected, Figure 16 reveals that to obtain a membrane with smaller pores, the fabrication process must be conducted under IRH conditions.

After immersing the membranes in water, a significant reduction in their effective pore size can be achieved compared to their initial size. This indicates that the nanofibers must first undergo a swelling stage to ensure that the pore size is suitable for use in the remediation process.

The presence of IONPs in the composite acts as a swelling stabilizer. By controlling the amount of IONPs, membranes could be tailored to the specific water contaminants to be filtered.

## 4. Conclusions

Humidity control in the electrospinning chamber is a critical factor for determining the final membrane properties, especially for those made from PVA and PVA + IONPs. Our results highlight the sensitivity of the electrospinning process to humidity and the effects of nanoparticle incorporation. Specifically, maintaining an intermediate humidity level results in nanofibers with a smaller average diameter compared to those processed at lower humidity levels. Unlike pure PVA, processing PVA + IONPs at low relative humidity leads to a bimodal diameter distribution, meaning that two different size ranges of nanofibers are present in the membrane. The presence of IONPs contributes to a larger effective pore radius, particularly when the manufacturing process is carried out at LRH. This underscores the importance of careful RH control in tailoring membrane properties, which could have implications for various applications such as filtration, drug delivery, or tissue engineering.

The swelling response of the PVA membrane differed from that of PVA + IONPs. This swelling process is not instantaneous; it has an associated rate, indicating that the increase in size or volume of the material when exposed to water occurs gradually over time. Swelling happens rapidly at the beginning of water passage through the membrane, suggesting that the fibers start with a minimum diameter and the swelling process initiates from this state. As water continues to pass through the membrane and the fibers increase in diameter, the average pore size between the fibers decreases. The reference to equilibrium suggests that a point exists where swelling stabilizes, achieving a balance between water absorption and fiber structural changes. IONPs seem to contribute to stabilizing the swelling of the composite material. Swelling stabilization is crucial for maintaining the structural integrity of the membrane during water filtration processes, preventing unwanted changes in the volume or shape of the material.

This research concludes that the addition of IONPs to a hydrophilic electrospun membrane alters its flow dynamics, which is influenced by its swelling properties and initial morphology controlled by the RH manufacturing condition. This type of research is essential for advancing water purification technologies and addressing specific water quality challenges.

## Figures and Tables

**Figure 1 membranes-14-00189-f001:**
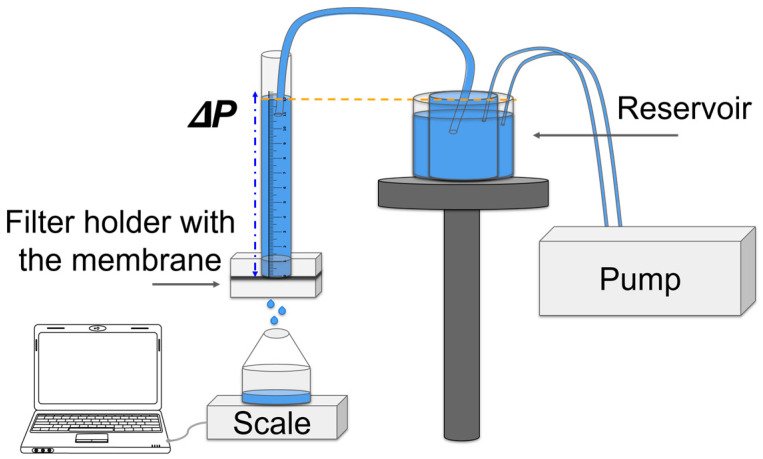
Permeate flow measurement setup for membrane filtration over time.

**Figure 2 membranes-14-00189-f002:**
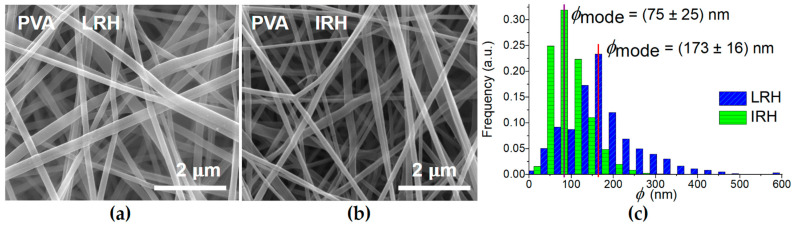
(**a**,**b**) SEM micrographs of electrospun PVA mats for both humidity conditions. (**c**) Influence of RH on diameter (ϕ) distribution.

**Figure 3 membranes-14-00189-f003:**
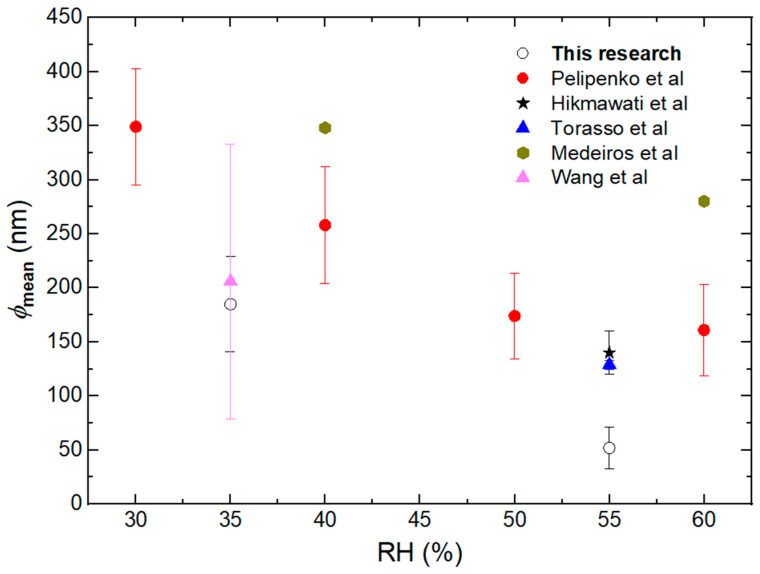
Mean diameter (ϕ_mean_) of PVA nanofibers at different RH obtained from the literature [12,26,27,29,40].

**Figure 4 membranes-14-00189-f004:**
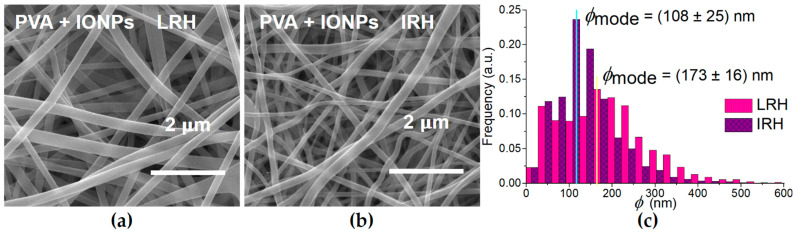
(**a**,**b**) SEM micrographs of electrospun PVA + IONPs mats for both relative humidity (RH) conditions. (**c**) Comparison of diameter (ϕ) distribution of PVA + IONPs mats.

**Figure 5 membranes-14-00189-f005:**
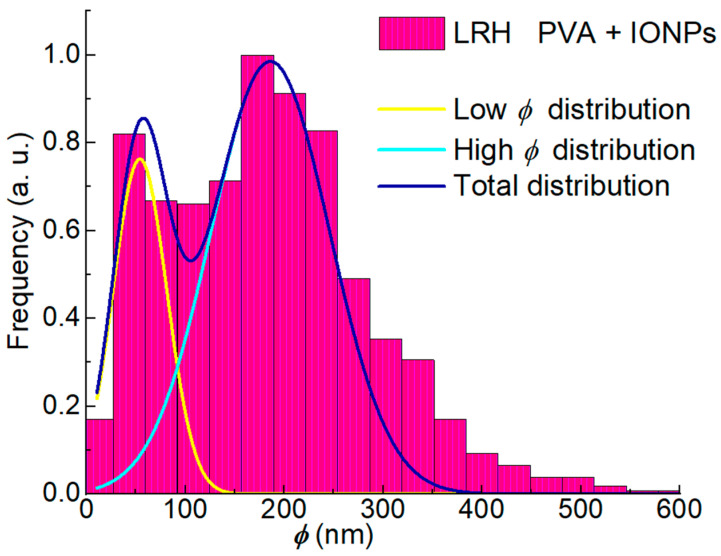
ϕ distribution of LRH PVA + IONPs mats considering two Gaussian distributions.

**Figure 6 membranes-14-00189-f006:**
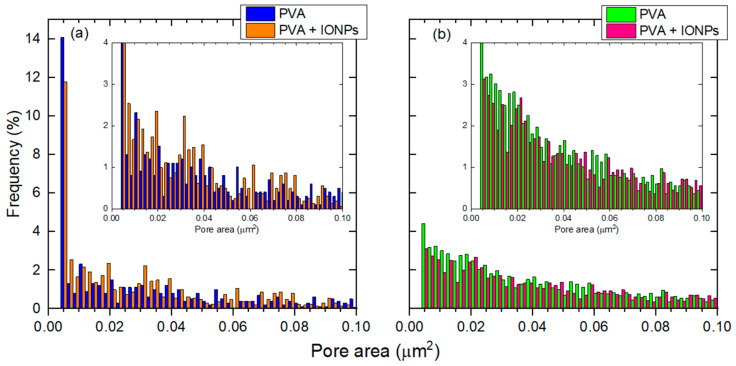
Pore area distribution for both materials separated by different RH conditions: (**a**) LRH and (**b**) IRH. The insets show details of each graph.

**Figure 7 membranes-14-00189-f007:**
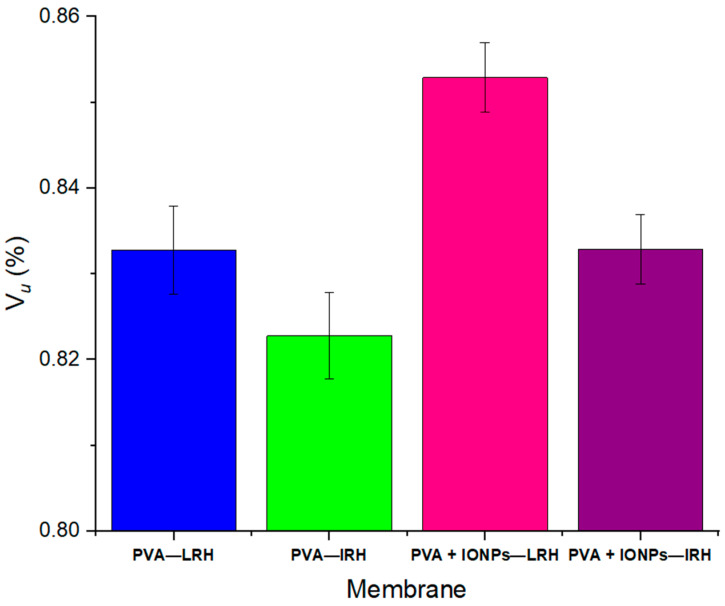
Porosity of the membranes.

**Figure 8 membranes-14-00189-f008:**
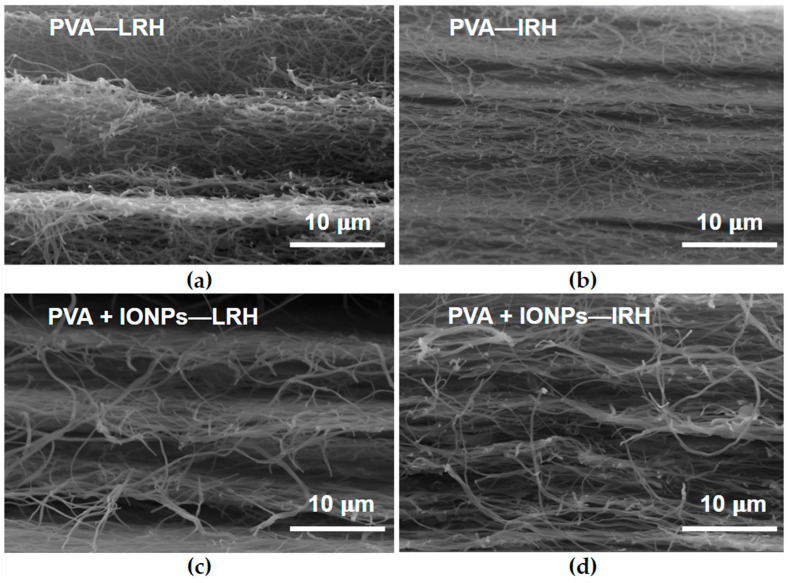
SEM micrographs of fractured electrospun mats, where (**a**) is for PVA at LRH, (**b**) is for PVA at IRH, (**c**) is for PVA + IONPs at LRH and (**d**) is for PVA + IONPs at IRH.

**Figure 9 membranes-14-00189-f009:**
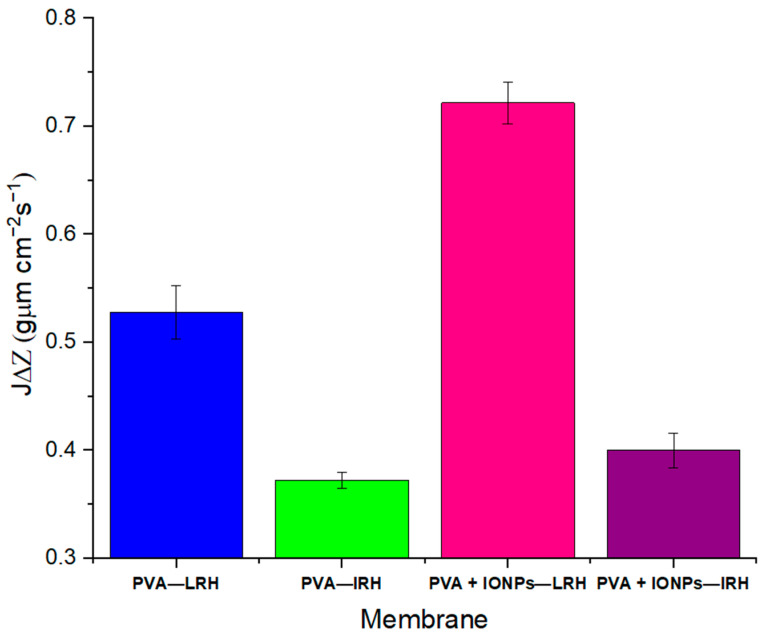
Toluene flow through PVA and PVA + IONPs membranes at LRH and IRH.

**Figure 10 membranes-14-00189-f010:**
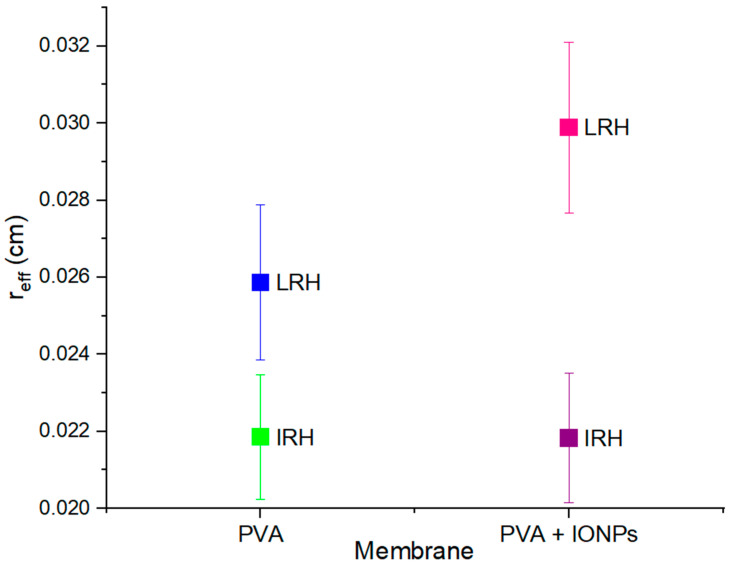
Effective radius (*r_eff_*) of the membrane pores in initial state, without swelling.

**Figure 11 membranes-14-00189-f011:**
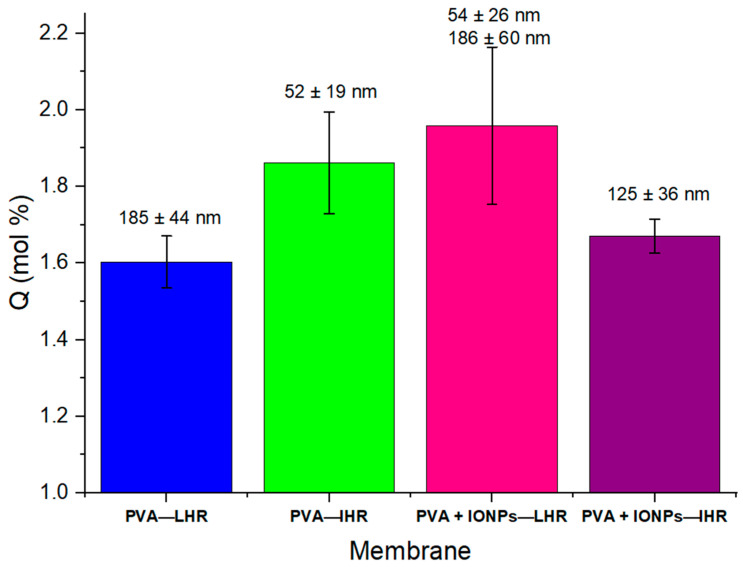
Mole percent pure water uptake of different membranes at room temperature (the mean nanofiber diameter are labeled for each membrane).

**Figure 12 membranes-14-00189-f012:**
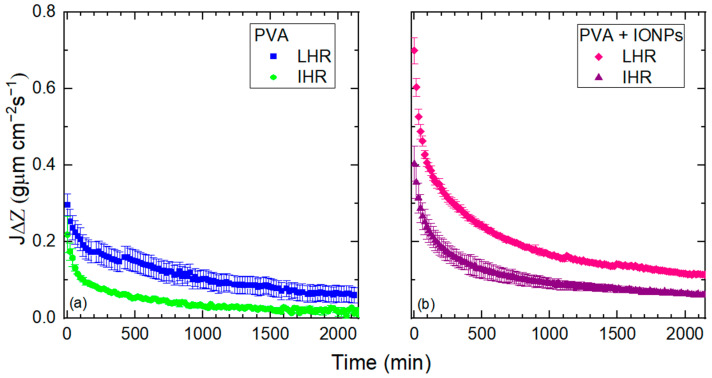
Permeate pure water flow for PVA (**a**) and PVA + IONPs (**b**) mats at constant pressure.

**Figure 13 membranes-14-00189-f013:**
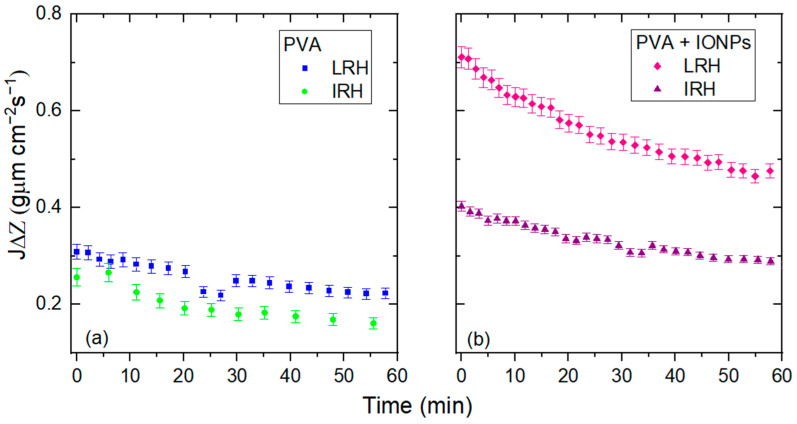
Pure water flow for PVA (**a**) and PVA + IONPs (**b**) mats at LRH and IRH.

**Figure 14 membranes-14-00189-f014:**
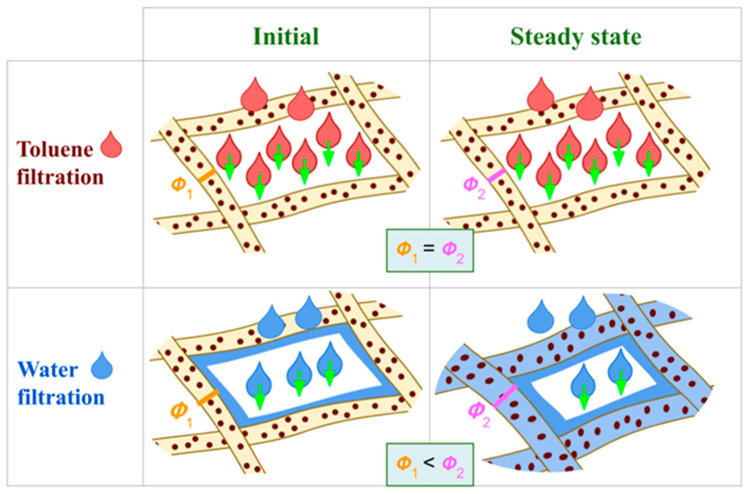
Filtration scheme between the nanofibers of PVA and PVA + IONPs membranes.

**Figure 15 membranes-14-00189-f015:**
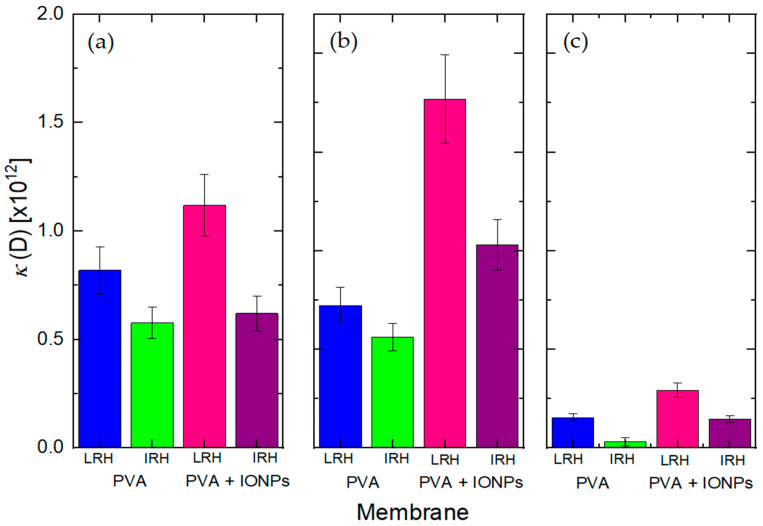
Permeability (*κ*) of PVA and PVA + IONPs membranes at LRH and IRH using (**a**) toluene, (**b**) distilled water (initial), and (**c**) distilled water (final).

**Figure 16 membranes-14-00189-f016:**
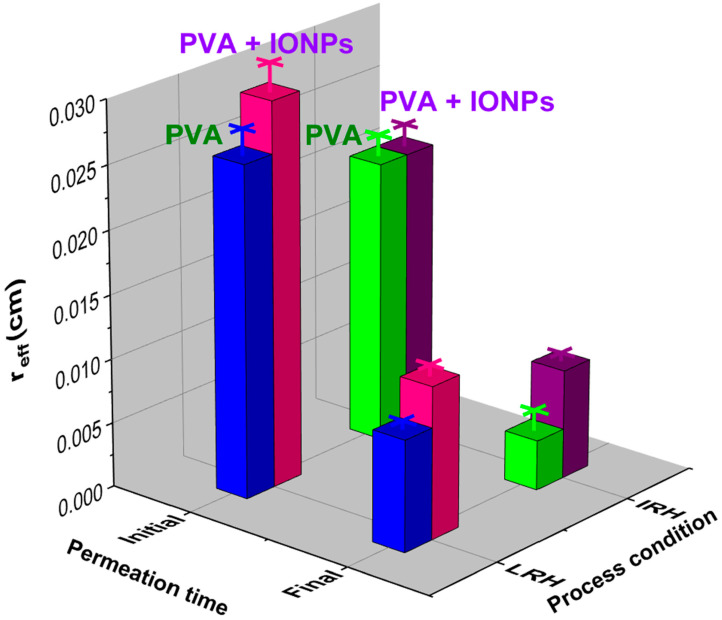
Changes in the effective radius (*r_eff_*) of the membrane pores with the type of membrane, process condition, and time of the permeation test.

**Table 1 membranes-14-00189-t001:** Composition and physical parameters of the PVA solutions.

Electrospinnable Solution	Viscosity [cp]	Conductivity [μS/cm]	pH	Surface Tension [mN/m]
PVA	309.33 ± 0.65	635.98 ± 0.31	5.331 ± 0.010	66.61 ± 0.18
PVA + IONPs	877.0 ± 2.2	2667.3 ± 1.7	2.498 ± 0.010	65.16 ± 0.38

**Table 2 membranes-14-00189-t002:** Statistical parameters of the distribution of nanofibers diameters, for the PVA and PVA + IONPs mats.

		PVA	PVA + IONPs	
	**LRH**	**IRH**	**LRH**	**IRH**
ϕ_mode_ (nm)	173	75	173	108
ϕ_mean_ (nm)	181	90	54 186	135
st. dev. (σ) (nm)	44	19	26 60	36

## Data Availability

The original contributions presented in the study are included in the article and Appendix A, further inquiries can be directed to the corresponding author.

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
