# Peer review of "Flow Dynamics through a High Swelling Nanofiber Membrane Processed at Different Relative Humidities: A Study on a FexOy/Polyvinyl Alcohol Composite"

_membranes, 2024, doi:10.3390/membranes14090189_

Round 1
Reviewer 1 Report
Comments and Suggestions for Authors
Recommendation: Major Revision
1. What is the meaning of IONPs in the abstract? Before presenting an abbreviation, it should be defined first.
2. Why did the author choose PVA for this study as PVA is a water-soluble polymer?
3. How did the authors prepare IONPs?
4. Why the content of IONPs was fixed at 1 % with respect to PVA?
5. The NPs are not visible in the PVA NFs, why?
6. The reason behind the change in morphology (nanofibers diameter) after the addition of IONPs should be explained.
7. The morphology of pristine IONPs should also be shown. Their shape and size should also be explained.
8. Some other characterization, such as XRD and FTIR are required to confirm the IONPs in the nanofibers.
9. The water contact angle should also be reported.
Author Response
1. What is the meaning of IONPs in the abstract? Before presenting an abbreviation, it should be defined first.
Thank you for bringing this error to our attention. In the revised version, we have added the definition of IONPs in the Abstract (lines 16-17).
2. Why did the author choose PVA for this study as PVA is a water-soluble polymer?
Thank you for pointing out that we were not clear enough about our choice of PVA. Several previous studies have demonstrated that although PVA is soluble in water and its nanofiber mat is similarly soluble, it is possible to convert it into a water-insoluble mat while retaining high hydrophilicity through heat treatment (HT) (Cimadoro 2018, Cimadoro 2020, Vergara-Rubio 2022). This approach maintains the advantages of electrospinning an aqueous solution—specifically, avoiding the use of organic solvents—while producing a membrane that, after HT, can be utilized for water remediation. PVA is chosen because once the membrane is insolubilized, the polymer has a high swelling capacity in water (Cimadoro 2020), which allows one to clearly observe how this effect influences the water permeation dynamics during the first hours.
The new version of the manuscript clarifies the concept in the Abstract (lines 15), Introduction (lines 32-41 and 98-100) and the Materials and Methods section (lines 144-146), written in red.
3. How did the authors prepare IONPs?
Since the synthesis method is the same as in the reference Torasso 2023, we have mentioned a brief description of it in the Materials and Methods section (lines 113-118) in the new version of the manuscript.
4. Why the content of IONPs was fixed at 1 % with respect to PVA?
The 1% was chosen because a previous study by our group demonstrated that this concentration produces an electrospun mat with excellent dispersion of IONPs and a higher concentration tends to cause agglomeration (Torasso et al. 2023).
5. The NPs are not visible in the PVA NFs, why?
We may not have been clear enough on this point, as the reviewer pointed out.
In this study, although we provide scanning electron microscopy (SEM) images, the IONPs are not visible due to their effective dispersion within the nanofibers and their size being smaller than the nanofiber diameter. However, in a recent study by our group, Torasso et al. (2023) utilized different characterization techniques on a similar material, demonstrating the presence of IONPs within the membrane nanofiber.
Transmission electron microscopy (TEM), Fourier transform infrared spectroscopy (FTIR) and X-ray diffraction (XRD) studies were performed to clarify this point and to show that the IONPs are dispersed within the nanofibers of the materials developed in our research.This information and discussion has been included in the Supplementary Material. The focus of this work is to study the flow dynamics in a polymeric membrane and how the presence of nanoparticles within the nanofibers affects the dynamics.
We have incorporated a reference in the manuscript in the Materials and Methods section (lines 155-159) and Results and Discussion section (lines 292-293).
6. The reason behind the change in morphology (nanofibers diameter) after the addition of IONPs should be explained.
We have analyzed this point in depth and we have rewritten how the addition of IONPs affects the morphology of the obtained membrane (see Results and Discussion section, lines 334-362).
7. The morphology of pristine IONPs should also be shown. Their shape and size should also be explained.
We appreciate your comments. Transmission electron microscopy images show that the IONPs, with an average size of (7.8 ± 5.3) nm, are dispersed within the nanofibers rather than on the surface (see Figure S1 SM). These results are consistent with previous reports from our research group for electrospun mats of similar composites (Torasso et al. 2023).
We have added a phrase in the Materials and Methods section (lines 155-159) and Results and Discussion section (lines 292-293) of the new manuscript version.
8. Some other characterization, such as XRD and FTIR are required to confirm the IONPs in the nanofibers.
We thank the reviewer for his suggestion. XRD and FTR studies confirming the presence of IONPs in the nanofibers have been included in the Supplementary Material, as well as Transmission Electron Microscopy images. We have added a phrase in the Materials and Methods section (lines 155-159) and Results and Discussion section (lines 292-293) of the new manuscript version.
9. The water contact angle should also be reported.
The material under investigation is a composite of polymeric nanofibers and air. The aim of this research is to investigate how the morphology of the membrane influences the flow dynamics. This morphology changes due to the swelling of the nanofibers when in contact with water, and these changes vary depending on the presence or absence of IONPs. Therefore, it is important to know the water contact angle for each nanofiber material. To achieve this, continuous films of these materials were prepared and used for water contact angle measurements as described in the Materials and Methods section (lines 214-220). Measuring the water contact angle directly on the electrospun membranes would not be accurate because a membrane is a composite of air pores and a specific material, meaning that the water contact angle is highly dependent on its morphology.
In materials like PVA, which have nanofibers that rapidly absorb water, the morphology of the membrane can change quickly due to swelling, leading to the closing of pores. This rapid change in morphology makes it difficult to obtain an accurate measurement of the water contact angle, as the surface properties of the membrane are constantly altering during the measurement process, resulting in large errors. We understand that this point was not clear in the manuscript and therefore we add a paragraph in the Materials and Methods section, lines 210-215.
Although, as the reviewer requests, this was also made and the results are shown in Table 1. The measurements made at ten different points of each membrane, at 5 seconds of drop contact, are shown in the next table. As can be seen, these values have an uncertainty between 45 and 77%. Although there seems to be a tendency towards a higher contact angle for the PVA+IONPs, this fact is not representative because there are no significant differences between the composite and the PVA electrospun mat.
Tabla 1: Water contact angle in the studied membranes.
|
PVA-BH |
(9±7) ° |
|
PVA-AH |
(12±8) ° |
|
PVA+NPs-BH |
(17±8) ° |
|
PVA+NPs-AH |
(20±9) ° |
In the case of our membranes, the morphology changes as a consequence of the swelling, and therefore the water contact angle of the membrane is only meaningful in the steady state, where it does not show significant differences in the case of PVA and PVA+IONPs.

Reviewer 2 Report
Comments and Suggestions for Authors
This study investigated the effects of relative humidity (RH) during electrospinning on the morphology and filtration dynamics of polyvinyl alcohol (PVA) and PVA-nanoparticle (PVA+IONPs) composite membranes. The impact of RH on fiber diameter, porosity, and permeability for contaminant retention capabilities were conducted. The results declared that PVA+IONPs membranes produced under IRH exhibit improved contaminant retention due to decreased permeability and altered fiber morphology. The manuscript is well-organized and make a good contribution for developing novel electrospun membranes for water treatment. Nevertheless, the work is till existing many issues. So, this work is suggested for publication in Membrane with major revisions. The main comments are listed below.
1. The composition of PVA+IONPs composite membranes need to be optimized for achieving the best contaminant retention capabilities.
2. The pure water flux and filtration efficiency should be provided in the revised manuscript.
3. The long-term duration test and recycling test should be conducted further to confirm the membrane performance for water treatment.
4. The differences in viscosity and conductivity between PVA and PVA+IONPs solutions are significant. How do these differences affect the electrospinning process beyond what is mentioned?
5. Recent reports on advanced organic/inorganic hybrid membranes and electrospun nanofiber membranes are recommended for improving the manuscript.
6. How uniformly are the IONPs distributed within the PVA fibers?
Comments on the Quality of English LanguageModerate editing of English language required.
Author Response
Reviewer #2
- The composition of PVA+IONPs composite membranes need to be optimized for achieving the best contaminant retention capabilities.
The objective of this study is not to investigate contaminant retention, but rather to evaluate the effect of the following factors on water permeation dynamics :
- a) The relative humidity (RH) in the electrospinning chamber and its effect on the morphology of the resulting membrane.
- b) The change in membrane morphology during the first hours of permeation due to swelling in water.
- c) The influence of the effects described in (a) and (b) on water permeation dynamics when the membrane is a PVA nanocomposite instead of pure PVA. For this evaluation, we used a membrane made from a previously studied nanocomposite, referred to as PVA+IONPs, as an example.
This represents a preliminary investigation into the membrane's response to water flow, with the objective of developing a future application for arsenic removal filters. Our group has previously conducted research on PVA membranes with varying concentrations of IONPs for arsenic removal under batch conditions.
The results presented in this work are crucial for the next stage, where adsorption tests of various contaminants in flow assays will be conducted to evaluate the efficiency of the membrane in removing them, with a particular focus on arsenic.
- The pure water flux and filtration efficiency should be provided in the revised manuscript.
We thank the reviewer for making us understand that we may have not been clear
In this study, we examined the permeation dynamics of two distinct flows: one comprising pure water (distilled water) and another containing toluene.
Several subtitles and text in the manuscript were changed to "pure water flux" to clarify this idea.
On the other hand, our research does not yet focus on filtration efficiency. Instead, it explores how membrane morphology influences flow dynamics and examines the morphological changes that occur during the initial hours of permeation, which are the central aspects of our study. As the referee rightly points out, future tests using contaminated water will enable the evaluation of filtration efficiency.
- The long-term duration test and recycling test should be conducted further to confirm the membrane performance for water treatment.
Thank you for your opinion on this point, as it has helped us to clarify the results of the flow dynamics curves shown in Figure 12. As can be seen in Figure 12, the permeate flow reaches a steady state (plateau). At this point, the pore size is constant and does not change over time, behaving as a traditional membrane under pure water flow. To further clarify this concept, we have added a phrase in the Results and discussion section (lines 488-494).
We acknowledge the reviewer's concern regarding the reuse of these membranes. This is obviously necessary when considering the contaminant removal. Our research group has experience in conducting similar research, and we have consistently engaged in the development of reuse strategies, as evidenced by our published works:
- Torasso, N.; Vergara-Rubio, A.; Pereira, R.; Martinez-Sabando, J.; Baudrit, J.R.V.; Cerveny, S.; Goyanes, S. An in Situ Approach to Entrap Ultra-Small Iron Oxide Nanoparticles inside Hydrophilic Electrospun Nanofibers with High Arsenic Adsorption. Chemical Engineering Journal 2023, 454, 140168, doi:10.1016/J.CEJ.2022.140168.
- Cimadoro, J.; Goyanes, S. Reversible Swelling as a Strategy in the Development of Smart Membranes from Electrospun Polyvinyl Alcohol Nanofiber Mats. Journal of Polymer Science 2020, 58, 737–746, doi:10.1002/POL.20190156.
- Coin, F.; Rodríguez-Ramírez, C.A.; Oyarbide, F.S.; Picón, D.; Goyanes, S.; Cerveny, S. Efficient Antibiotic Removal from Water Using Europium-Doped Poly(Vinyl Alcohol) Nanofiber Mats Esterified with Citric Acid. Journal of Water Process Engineering 2024, 63, 105447, doi:10.1016/J.JWPE.2024.105447.
- Vergara-Rubio, A.; Ribba, L.; Picón, D.; Candal, R.; Goyanes, S. A Highly Efficient Nanostructured Sorbent of Sulfuric Acid from Ecofriendly Electrospun Poly(Vinyl Alcohol) Mats. Industrial and Engineering Chemistry Research 2022, 61, 2091–2099, doi:10.1021/ACS.IECR.1C03530.
- Picón, D.; Vergara-Rubio, A.; Estevez-Areco, S.; Cerveny, S.; Goyanes, S. Adsorption of Methylene Blue and Tetracycline by Zeolites Immobilized on a PBAT Electrospun Membrane. Molecules 2023, 28, 81. https://doi.org/10.3390/molecules28010081
- Pereira, P.P., Fernandez, M., Cimadoro, J. et al. Biohybrid membranes for effective bacterial vehiculation and simultaneous removal of hexavalent chromium (CrVI) and phenol. Appl Microbiol Biotechnol 105, 827–838 (2021). https://doi.org/10.1007/s00253-020-11031-x
Despite the fact that we have these previous studies, we have not addressed the effect of relative humidity in the electrospinning chamber on the morphology of the resulting membrane until the present research. Most of our previous studies have been conducted in batch conditions, as there has been no prior investigation into the flow dynamics of the membranes. This study demonstrates the important role of the initial permeation day in the context of employing PVA membranes as an exclusion filter based on size.
- The differences in viscosity and conductivity between PVA and PVA+IONPs solutions are significant. How do these differences affect the electrospinning process beyond what is mentioned?
In the new version of the manuscript, we have analyzed this point in depth and we have rewritten how these differences affect the electrospinning process and therefore the membrane morphology obtained (see Results and discussion section, lines 334-362).
- Recent reports on advanced organic/inorganic hybrid membranes and electrospun nanofiber membranes are recommended for improving the manuscript.
Thank you for your suggestion. We have added new references that improve the manuscript. The Introduction section (lines 48-50) and the Results and Discussion section (lines 352-354) have been updated.
- How uniformly are the IONPs distributed within the PVA fibers?
Considering the reviewer's comment, we have included a supplementary material with TEM images, XRD patterns, and FTIR spectra to show the good dispersion of the IONPs inside the PVA nanofiber. These techniques have been introduced in the Materials and Methods section (lines 155-159) and a sentence on this point has been included in the Results and Discussion section (lines 292-293).

Round 2
Reviewer 1 Report
Comments and Suggestions for Authors
The authors have addressed all the concerns raised in the previous round of reviews. However, I noticed mistakes in the revised manuscript regarding the citation of literature. Please use the same citation style as suggested by the journal. The paper can be accepted after addressing this issue.
Author Response
Thank you for your suggestions. We have made the revisions accordingly.
Reviewer 2 Report
Comments and Suggestions for Authors
The quality of the manuscript has been enhanced according to the reviewers' comments, and it could be accepted now.
Author Response
Thank you for your time and valuable suggestion.